# Effects of Red and Blue Light on the Growth, Photosynthesis, and Subsequent Growth under Fluctuating Light of Cucumber Seedlings

**DOI:** 10.3390/plants13121668

**Published:** 2024-06-16

**Authors:** Tengqi Wang, Qiying Sun, Yinjian Zheng, Yaliang Xu, Binbin Liu, Qingming Li

**Affiliations:** 1College of Horticulture Science and Engineering, Shandong Agriculture University, Tai’an 271018, China; wang_tengqi@163.com (T.W.); sqy1049698466@163.com (Q.S.); 2Institute of Urban Agriculture, Chinese Academy of Agricultural Sciences, Chengdu 610299, China; zhengyinjian@caas.cn (Y.Z.); xuyaliang@caas.cn (Y.X.); 3College of Food and Biological Engineering, Chengdu University, Chengdu 610106, China

**Keywords:** cucumber, red light, blue light, steady-state photosynthesis, dynamic photosynthesis, fluctuating light

## Abstract

The effects of red and blue light on growth and steady-state photosynthesis have been widely studied, but there are few studies focusing on dynamic photosynthesis and the effects of LED pre-treatment on cucumber seedlings’ growth, so in this study, cucumber (*Cucumis sativus* L. cv. Jinyou 365) was chosen as the test material. White light (W), monochromatic red light (R), monochromatic blue light (B), and mixed red and blue lights with different red-to-blue ratios (9:1, 7:3, 5:5, 3:7, and 1:9) were set to explore the effects of red and blue light on cucumber seedlings’ growth, steady-state photosynthesis, dynamic photosynthesis, and subsequent growth under fluctuating light. The results showed that compared with R and B, mixed red and blue light was more suitable for cucumber seedlings’ growth, and the increased blue light ratios would decrease the biomass of cucumber seedlings under mixed red and blue light; cucumber seedlings under 90% red and 10% blue mixed light (9R1B) grew better than other treatments. For steady-state photosynthesis, blue light decreased the actual net photosynthetic rate but increased the maximum photosynthetic capacity by promoting stomatal development and opening; 9R1B exhibited higher actual net photosynthetic rate, but the maximum photosynthetic capacity was low. For dynamic photosynthesis, the induction rate of photosynthetic rate and stomatal conductance were also accelerated by blue light. For subsequent growth under fluctuating light, higher maximum photosynthetic capacity and photoinduction rate could not promote the growth of cucumber seedlings under subsequent fluctuating light, while seedlings pre-treated with 9R1B and B grew better under subsequent fluctuating light due to the high plant height and leaf area. Overall, cucumber seedlings treated with 9R1B exhibited the highest biomass and it grew better under subsequent fluctuating light due to the higher actual net photosynthetic rate, plant height, and leaf area.

## 1. Introduction

As a popular vegetable, cucumber is widely cultured around the world, and cucumber seedlings are mainly produced in controlled environments [1]; using a light control strategy during production could promote the growth of seedlings [2]. Compared with other light qualities, including yellow and green, red and blue light are more suitable for plant growth [3]. Although lots of studies have explored the effects of red and blue light on the growth and photosynthesis of plants, the studies focusing on the dynamic photosynthesis and subsequent growth of cucumber are fewer.

Red and blue light could significantly influence plant growth. Previous studies have shown that cucumber [4], tomato [5], and lettuce [6] grow better under mixed red and blue light, and changes in the red-to-blue ratios could significantly affect the biomass and morphology of plants [5,7]. In general, under mixed red and blue light, increasing the blue light ratios could reduce plant height, leaf area, and shoot dry weight, while increasing stem diameter and root dry weight to make seedlings more compact. These results have been confirmed in cucumber [4], tomato [1], and soybean seedlings [8]. Although lots of studies have explored the effects of red and blue light on the growth of cucumber seedlings, there were few studies focused on the correlation between blue light ratio and related growth parameters of cucumber seedlings.

Previous studies have mainly focused on steady-state photosynthesis, which represents the photosynthetic capacity of plants under specific environmental conditions [9,10]. However, steady-state photosynthesis cannot fully reflect the actual photosynthetic performance of plants grown in unstable environments [11]. Therefore, more studies pay attention to photosynthesis in unstable environments [12], especially under fluctuating light [13,14]. For steady-state photosynthesis, lots of studies have demonstrated that red and blue light can more effectively activate photosynthesis [15], and compared with red light, blue light plays a more important role in promoting the development of photosynthetic organs and maintaining the operation of photosynthesis [16,17]. Blue light can significantly improve the maximum photosynthetic capacity of plants by improving stomatal conductance [18], photosystem activity [19], and Calvin cycle-related key enzyme activity and gene expression [20]. Compared with monochrome red light or monochrome blue light, the photosynthetic capacity of cucumber, tomato, and soybean is significantly improved under mixed red and blue light, and photosynthesis can be further optimized by optimizing the red-to-blue ratio [21]. For dynamic photosynthesis, it has been illustrated that variety [10], nitrogen content [22], and water management strategy [23] can affect the dynamic photosynthesis of rice and canola seedlings. However, the effect of red and blue light on dynamic photosynthesis has not been fully explored; although previous studies have found that blue light can accelerate the photoinduction rate of cucumber seedlings [24], the mechanism of blue light affecting dynamic photosynthesis is still unclear.

Red and blue light can not only affect plant growth and photosynthesis, but also affect the subsequent growth of plants. Previous studies have shown that lemon balm and lettuce pre-treated with mixed red and blue light grow better under subsequent drought stress [25,26]. In addition, pre-treatment with blue light at the seedling stage could promote the growth of cucumber seedlings after transplanting them to natural light [27]. These results indicate that pre-treatment with red and blue light could affect the subsequent growth of plants. During plant growth, it is difficult to maintain constant light intensity on the surface of leaves [11]. Thus, it is valuable to find new ways to improve the ability of plants to cope with fluctuating light. A previous study has indicated that enhancing stomatal kinetics by optogenetic manipulation could effectively accelerate stomatal opening and closing, thus increasing the biomass of Arabidopsis under fluctuating light [28]. In addition, a previous study found that red and blue light could affect the induction rate of photosynthetic rate and stomatal conductance [24]. Therefore, it is necessary to explore whether pre-treated with red and blue light can affect the growth of cucumber seedlings under subsequent fluctuating light.

In this study, white light, monochrome red light, monochrome blue light, and mixed red and blue lights with different red-to-blue ratios were set to investigate: (1) the effects of blue light ratios on biomass and morphological characteristics of cucumber seedlings, (2) the effects of red and blue light on steady-state and dynamic photosynthesis of cucumber seedlings, and (3) the effects of red and blue light on the growth of cucumber seedlings under subsequent fluctuating light. This study aims to provide a theoretical basis for the optimization of red-to-blue ratio during cucumber seedling cultivation.

## 2. Results

### 2.1. Effect of Red and Blue Light on Growth and Morphology of Cucumber Seedlings

The morphological characteristics of cucumber seedlings are shown in Appendix A, and related parameters are shown in Figure 1. After 9 d of treatment, compared with white light (W), plant height and stem diameter under monochrome red light (R) were significantly increased by 41.4% and 10.2%, respectively. Although the change in leaf area was not obvious under R, the dry weight and fresh weight of leaf, shoot, root, and whole plant, Specific leaf weight, root–shoot ratio, and seedling index were significantly reduced. Specific leaf weight reached the lowest value under R. Similar to R, monochrome blue light (B) significantly increased plant height and stem diameter, by 198.8% and 23.4% over W, and plant height and stem diameter reached the highest values under B. Meanwhile, leaf area, fresh weight, and dry weight of leaf and root were significantly reduced by B, but the stem fresh weight was significantly increased and reached the highest value. The fresh weight of the shoot and whole plant under B were higher than those under W, increasing by 13.2% and 7.0%, respectively. However, the shoot dry weight and whole plant dry weight under B were lower than those under W. In addition, B significantly reduced specific leaf weight, root–shoot ratio, and seedling index by 15.4%, 36.6%, and 53.8%, respectively, over W. Under R, the whole plant dry weight and specific leaf weight reached the lowest values, while under B, the root–shoot ratio and seedling index reached the lowest values. Under mixed red and blue light, an increase of blue light ratio increased specific leaf weight but reduced biomass; specific leaf weight reached the lowest values under 90% red and 10% blue mixed light (9R1B), but leaf area, leaf fresh and dry weight, shoot fresh and dry weight, whole plant fresh and dry weight, root–shoot ratio, and seedling index reached the highest values.

As shown in Figure 2, under mixed red and blue light, specific leaf weight was positively correlated with blue light ratio, while other growth indicators were negatively correlated with blue light ratio. Except for specific leaf weight, other growth parameters showed a positive correlation, indicating that under mixed red and blue light, an increase of blue light ratio could significantly increase specific leaf weight, but reduce plant height, stem diameter, leaf area, fresh weight, and dry weight.

In summary, R and B could inhibit the growth of cucumber seedlings, and the inhibition effect of R was more obvious than that of B. Cucumber seedlings exhibited the best growth condition under 9R1B treatment.

### 2.2. Effect of Red and Blue Light on the Actual Net Photosynthetic Rate of Cucumber Seedlings

As shown in Figure 3, compared with W, actual net photosynthetic rates (A_A_) under R and B were decreased by 12.4% and 20.9% respectively: A_A_ was slightly decreased under 70% red and 30% blue mixed light (7R3B) and 50% red and 50% blue mixed light (5R5B), while significantly reduced under 30% red and 70% blue mixed light (3R7B) and 10% red and 90% blue mixed light (1R9B) by 26.8% and 23.2% over W, respectively. Red light could effectively improve A_A_: it reached the lowest value under 1R9B and reached the highest under 9R1B. Actual intercellular carbon dioxide concentration (Ci_A_) was significantly decreased under R, but changed slightly under other treatments. Actual stomatal conductance (Gs_A_) was significantly decreased under R, by 42.5% over W, but increased by blue light; it reached the highest value while blue light ratio was 50%. R significantly increased actual instantaneous water use efficiency (iWUE_A_) by 52.2% over W. Blue light could decrease iWUE_A_: under 3R7B and B, it was significantly decreased by 40.6% and 32.9%, respectively. iWUE_A_ reached the lowest value under 3R7B and reached the highest under R, followed by 9R1B. In summary, red light could improve A_A_ and iWUE_A_, but reduce Ci_A_ and Gs_A_, while the effect of blue light was opposite to that of red light.

### 2.3. Effect of Red and Blue Light on Steady-State Photosynthesis of Cucumber Seedlings

The various tendencies of photosynthetic gas exchange parameters during light induction are shown in Figure 4, and the steady-state photosynthetic parameters during light induction are shown in Table 1. Compared with W, R and 9R1B significantly decreased net photosynthetic rate after light induction (A_f_), decreasing by 43.7% and 26.4%, respectively. 7R3B, 5R5B, 3R7B, and 1R9B significantly increased A_f_ by 22.5%, 31.4%, 32.8%, and 22.4% over W, respectively, indicating that blue light could significantly increase A_f_; it reached the lowest value under R and reached the highest under 3R7B. Stomatal conductance after light induction (Gs_f_) decreased significantly under R, by 27.6% more compared to W, except for R and 9R1B. Gs_f_ under other treatments was higher than under W, but the differences were not significant. Intercellular carbon dioxide concentration after light induction (Ci_f_) was significantly increased under R and 9R1B, and it was the highest under R. Instantaneous water use efficiency after light induction (iWUE_f_) was significantly decreased by 23.0% and 24.7% under R and 9R1B compared to W, respectively, iWUE_f_ was increased by blue light; it reached the highest value under 5R5B and reached the lowest under 9R1B. Dark respiration rate (R_d_) only decreased significantly under R by 61.5% compared to W. In summary, compared with red light, blue light could increase net photosynthetic rate, stomatal conductance, and instantaneous water use efficiency of cucumber seedlings under high light intensity (1500 μmol·m^−2^·s^−1^).

### 2.4. Effect of Red and Blue Light on Dynamic Photosynthesis of Cucumber Seedlings

The dynamic photosynthetic parameters are shown in Table 2. Compared with W, time to reach 50% of steady-state net photosynthetic rate (T_50%A_) was only significantly decreased by R. Compared with W, time to reach 90% of steady-state net photosynthetic rate (T_90%A_) was decreased under 5R5B, 3R7B, 1R9B, and B, but the difference was only significant under 1R9B, which decreased by 22.4% compared to W. T_90%A_ reached the lowest value under 1R9B and reached the highest under R, indicating blue light could effectively shorten total photoinduction time. The activation rate of Rubisco (1/τ_R_) was significantly increased under R and 7R3B by 241.6% and 83.0% compared to W, respectively; it reached the highest under R and reached the lowest under 9R1B. Time to reach 50% steady-state stomatal conductance (T_50%Gs_) changed slightly under different treatments. Compared with W, time to reach 90% steady-state stomatal conductance (T_90%Gs_) was decreased under 7R3B, 5R5B, 3R7B, 1R9B, and B, but only significantly decreased under 1R9B by 22.8% compared to W, indicating blue light could shorten the induction time of Gs. Time lag of Gs response (λ) was significantly increased under R by 38.9% compared to W, while 1R9B significantly decreased λ by 21.9%; it reached the highest value under R and reached the lowest under 1R9B, indicating blue light could accelerate the Gs response rate. Maximum stomatal conductance increase rate (Sl_max_) was decreased significantly under R by 31.3% compared to W, and was significantly increased under 5R5B and 3R7B by 54.7% and 56.5% compared to W, respectively, Sl_max_ reached the lowest under R and reached the highest value under 3R7B indicating blue light could increase the induction rate of Gs. Overall, blue light could increase the induction rate of A and Gs during light induction.

### 2.5. Effect of Red and Blue Light on Stomatal Characteristics of Cucumber Seedlings

Stomatal characteristics are shown in Appendix A and related parameters are shown in Figure 5. Compared with W, adaxial and abaxial stomatal density (SD) were significantly decreased under R by 27.1% and 16.9%, respectively. B significantly reduced abaxial SD by 20.4% compared to W, but had less effect on adaxial SD; under mixed red and blue light, adaxial and abaxial SD increased with the increase of blue light ratio. Adaxial and abaxial SD reached the highest value under 1R9B. R and B significantly increased adaxial epidermal cell density (ECD) by 12.8% and 8.6% compared to W, respectively. Under mixed red and blue light, adaxial ECD increased with the increase of blue light ratio. Adaxial ECD was the lowest under 9R1B and was the highest under 1R9B. The variation in epidermal cell area (ECA) was opposite to that in ECD. The variation of stomatal index (SI) was similar to that of SD. Adaxial and abaxial stomatal aperture area (AA) decreased significantly under R by 24.6% and 33.0% compared to W, respectively. Under mixed red and blue light, AA increased with the increase of blue light ratio. For stomatal aperture area per area (AAPA), compared with W, R significantly reduced adaxial and abaxial AAPA by 44.2% and 44.3%, respectively. Under mixed red and blue light, AAPA increased with the increase of blue light ratio. AAPA was the lowest under R, followed by 9R1B, and was the highest under 1R9B. Compared with W, R significantly reduced stomatal area (SA) and stomatal area per area (SAPA). Under mixed red and blue light, abaxial SA and SAPA increased with the increase of blue light ratio, while adaxial SA was significantly increased by 5R5B, and abaxial SAPA was significantly increased under 5R5B and 1R9B. Overall, blue light could promote the development and opening of stomata.

### 2.6. Effect of Red and Blue Light on the Growth of Cucumber Seedlings under Subsequent Fluctuating Light

As shown in Figure 6, cucumber seedlings pre-treated with R, 9R1B, and B exhibited higher plant height and stem diameter after treatment with fluctuating light for 15 d. The increase of blue light ratio under mixed red and blue light resulted in a decrease of plant height, but an increase of stem diameter. Leaf area was the highest under 9R1B after treatment with fluctuating light, while it decreased with the increase of blue light ratio under mixed red and blue light, but seedlings pre-treated with B exhibited higher leaf area. The variation of leaf fresh weight and leaf dry weight was similar to that of leaf area. After fluctuating light treatment, the shoot fresh weight and shoot dry weight increased significantly under 9R1B and decreased with the increase of blue light ratio under mixed red and blue light. Seedlings pre-treated with 9R1B and B exhibited higher root fresh weight and root dry weight; after treatment with fluctuating light for 15 d, root fresh weight and root dry weight reached their highest values under B, followed by 9R1B. Cucumber seedlings pre-treated with R, 9R1B, and B exhibited higher fresh weight and dry weight; the fresh weight and dry weight reached the highest value under B after treatment with fluctuating light for 15 d, followed by 9R1B. Cucumber seedlings pre-treated with 9R1B still exhibited the highest seedling index after treatment with fluctuating light for 15 d, followed by B, and the increase of blue light ratio led to a decrease of seedling index under mixed red and blue light. Overall, cucumber seedlings pre-treated with 9R1B and B grew better under fluctuating light.

As shown in Figure 7, the growth rate of plant height (GR_plant height_) was significantly increased by B, but it was significantly decreased under 5R5B, 3R7B, and 1R9B by 17.2%, 24.1% and 44.7% compared to W, respectively; it reached the lowest value under 1R9B and reached the highest value under B. Growth rate of stem diameter (GR_stem diameter_) was only significantly increased by 3R7B. Growth rate of leaf area (GR_leaf area_) significantly increased under R, 9R1B, and B, by 23.0%, 29.8%, and 28.2% compared to W, respectively, and it was significantly decreased by 5R5B, 3R7B, and 1R9B. GR_leaf area_ reached the lowest value under 1R9B and reached the highest value under 9R1B. Growth rate of leaf fresh weight (GR_leaf FW_) significantly increased under R, 9R1B, 7R3B, and B by 63.3%, 66.8%, 35.5%, and 82.5% compared to W, respectively, and it was significantly decreased by 1R9B. GR_leaf FW_ reached the lowest value under 1R9B and reached the highest value under B, followed by 9R1B. Growth rate of stem fresh weight (GR_stem FW_) was significantly increased under R, 9R1B, and B by 26.2%, 43.5%, and 65.2% compared to W, respectively, and it was significantly decreased under 1R9B; it was the lowest under 1R9B and was the highest under B, followed by 9R1B. Growth rate of root fresh weight (GR_root FW_) was significantly increased under R, 9R1B, 7R3B, and B by 110.4%, 101.4%, 80.6%, and 300.4% compared to W, respectively, and it was the lowest under 5R5B and was the highest under B, followed by R. The variation of growth rate of leaf dry weight (GR_leaf DW_) was similar to that of GR_leaf FW_, while the variation of growth rate of stem dry weight (GR_stem DW_) and root dry weight (GR_root DW_) were similar to those of GR_stem FW_ and GR_root FW_. The growth rate of total fresh weight (GR_total FW_) and total dry weight (GR_total DW_) varied similarly to those of leaves and stems. The variation of growth rate of seedling index (GR_seedling index_) was consistent with that of GR_root DW_; it was the highest under B, followed by R. Overall, pre-treatment with R, B, and 9R1B could improve the growth rate of cucumber seedlings under subsequent fluctuating light, and the effect of B was more obvious.

### 2.7. Correlation Analysis under Mixed Red and Blue Light

The correlation between plant growth and photosynthetic characteristics under mixed red and blue light is shown in Figure 8A. Leaf area, total DW, seedling index, A_A_, and iWUE_A_ were negatively correlated with blue light ratio (BL ratio), while Gs_A_, A_f_, GS_f_, and iWUE_f_ were positively correlated with BL ratio, indicating that the increase of blue light ratio under mixed red and blue light could decrease biomass, A_A_, and iWUE_A_, but increase Gs_A_, A_f_, GS_f_, and iWUE_f_. Leaf area, dry weight, and seedling index were positively correlated with A_A_ and iWUE_A_, but negatively correlated with Gs_A_, A_f_, Gs_f_, and iWUE_f_, indicating that a higher actual net photosynthetic rate could increase the biomass of cucumber seedlings. A_A_ was negatively correlated with Gs_A_ but positively correlated with iWUE_A_, indicating that A_A_ and iWUE_A_ could be improved together by optimizing the red-to-blue ratio. A_f_, Gs_f_, and iWUE_f_ were positively correlated with each other, indicating that A_f_, GS_f_, and iWUE_f_ could be simultaneously improved by optimizing the red-to-blue ratio.

The relationship between steady-state photosynthesis, dynamic photosynthesis, and stomatal characteristics is shown in Figure 8B. A_f_ and Gs_f_, stomatal density (SD), and stomatal aperture area per area (AAPA) were positively correlated with BL ratio, while T_A90%_, T_Gs50%_, and T_Gs90%_ were negatively correlated with BL ratio, indicating that the increase of blue light ratio under mixed red and blue light would increase A_f_ and Gs_f_, accelerate photoinduction, and promote stomatal development and opening. A_f_ and Gs_f_ were negatively correlated with T_A90%_, T_Gs50%_, and T_Gs90%_ and positively correlated with SD and AAPA, indicating that higher steady-state net photosynthetic rate and stomatal conductance could accelerate photoinduction. T_A50%_, T_A90%_, T_Gs50%_, and T_Gs90%_ were negatively correlated with SD and AAPA, indicating that blue light could accelerate photoinduction through promoting stomatal development.

As shown in Figure 8C, GR_leaf area_ and GR_total DW_ were significantly negatively correlated with BL ratio, indicating that the increase of blue light ratio could reduce the growth rate of cucumber seedlings under subsequent fluctuating light. GR_leaf area_ and GR_total DW_ were positively correlated with leaf area, total DW, and seedling index, indicating that cucumber seedlings with higher leaf area and dry weight exhibited higher growth rate under subsequent fluctuating light. GR_leaf area_ and GR_total DW_ were negatively correlated with A_f_ and Gs_f_, but positively correlated with T_A90%_, T_Gs50%_, and T_Gs90%_, indicating that compared with steady-state and dynamic photosynthesis, biomass and morphological characteristics exhibited a more obvious effect on the growth of cucumber seedlings under subsequent fluctuating light.

## 3. Discussion

### 3.1. Plant Growth and Morphological Characteristics of Cucumber Seedlings Were Significantly Affected by Red and Blue Light

Red and blue light could significantly affect the growth and morphology of plants. Previous studies have shown that red light and blue light play key roles in maintaining the normal growth and development of plants [15]. However, plants growing under monochrome red light or blue light usually exhibit higher plant height and lower biomass [6,29]. Compared with monochromatic red and blue light, plants grow better under mixed red and blue light, and changes to blue light ratio could significantly affect the biomass and morphology of plants [4,5]. In this experiment, compared with white light, the growth of cucumber seedlings under monochrome red light and monochrome blue light was obviously repressed, and dry weight reached the lowest values under monochrome red light (Figure 1). The result was similar to that obtained by Hernández et al. [5], who found that monochrome red light could inhibit the growth of cucumber and tomato. Plant height was the highest under monochrome blue light, but was decreased by blue light under mixed red and blue light (Figure 1 and Figure 2). This is mainly because blue light could only promote stem elongation under low phytochrome activity, so the effect of blue light on plant height was affected by red light [30]. Under mixed red and blue light, an increase of blue light ratio could significantly increase specific leaf weight, while other growth parameters were significantly decreased (Figure 2), indicating that blue light could effectively increase the leaf thickness of cucumber seedlings, but significantly decrease biomass. Previous studies have found that under mixed red and blue light, increasing blue light ratio could improve leaf thickness, stem diameter, and root dry weight to make plants more compact [4,29], but at same time, the negative effect of blue light on leaf area should not be ignored. It is important to find a suitable blue light ratio to fully amplify the positive effect of blue light. Cucumber seedlings grown under 9R1B exhibited the highest biomass and seedling index (Figure 1), indicating adding low-intensity blue light to red light could promote the growth of cucumber seedlings. This result was consistent with previous studies on tomato [31] and soybean [8].

### 3.2. Photosynthetic Characteristics of Cucumber Seedlings Were Significantly Affected by Red and Blue Light

Monochrome red light usually causes red light syndrome, resulting in a disorder of photosynthesis, but this effect could be alleviated by blue light [17,32,33], and the increase of blue light ratio could further improve the maximum photosynthetic capacity of most plants, including cucumber [4], tomato [34], and lettuce [21]. In this study, the effect of red and blue light on maximum photosynthetic capacity of cucumber seedlings was consistent with previous studies. Red light significantly reduced the maximum photosynthetic capacity of cucumber seedlings, while blue light could significantly improve maximum photosynthetic capacity by promoting stomatal development and opening (Figure 5). The effect of red and blue light on actual net photosynthetic rate of cucumber seedlings was different from maximum photosynthetic capacity. Actual net photosynthetic rate increased under red light but decreased under blue light (Figure 3), because the quantum utilization efficiency of red light is higher than that of blue light [35]. This also indicated that red light caused little restriction on photosynthesis under low light intensity. After analyzing the relationship between seedling growth, actual net photosynthetic rate and maximum photosynthetic capacity under mixed red and blue light, we found that the main factors affecting the biomass of cucumber seedlings were leaf area and actual net photosynthetic rate, but not maximum photosynthetic capacity (Figure 8A), indicating that red and blue light could affect biomass through affecting plant morphology and actual net photosynthetic rate. The key roles of plant morphology have also been emphasized in a previous study on rice plants [36].

In recent years, photosynthesis under fluctuating light has been widely studied, and the one-step light intensity change method has been widely used to evaluate the ability of plants to cope with fluctuating light [22,37]. Using one-step light intensity change method, we found that blue light could accelerate photoinduction, although red light could accelerate the initial photoinduction rate by increasing the activation rate of Rubisco, but the increase of stomatal conductance was too slow (Table 2), thus limiting the increase of net photosynthetic rate after the initial stage of photoinduction (Appendix A). This result was consistent with Li et al. [24], who has found that blue light could accelerate the increase rate of stomatal conductance. After analyzing the correlation between key steady-state photosynthetic parameters, dynamic photosynthetic parameters, and stomatal characteristics under mixed red and blue light, we found that an increase of blue light ratio could accelerate photoinduction rate, with an increase of stomatal density and stomatal conductance (Figure 8B) indicating blue light could accelerate photoinduction rate through promoting stomatal development, but the photoinduction was also affected by other factors, for example, mesophyll conductance and biochemical reactions [38,39].

In summary, red and blue light could significantly affect the steady-state and dynamic photosynthesis of cucumber seedlings. Red light played key roles in increasing the actual net photosynthetic rate, while blue light played key roles in increasing maximum photosynthetic capacity and photoinduction rate.

### 3.3. Growth of Cucumber Seedlings under Subsequent Fluctuating Light Was Significantly Affected by Red and Blue Light

Red and blue light not only could affect the physiological characteristics of plants, but also could affect the subsequent growth of plants. Previous studies have shown that lettuce and lemon balm pre-treated with mixed red and blue light grew better under subsequent drought stress [25,26]. In this study, cucumber seedlings pre-treated with monochrome red light, monochrome blue light and 9R1B grew better under subsequent fluctuating light, and the growth rate reached the highest value under monochrome blue light (Figure 6 and Figure 7). Although blue light could increase photoinduction rate, it could not promote the growth of cucumber seedlings under fluctuating light, while the higher leaf area and biomass before fluctuating light treatment could significantly improve the growth rate of cucumber seedlings under fluctuating light (Figure 8C), indicating that the leaf area and dry weight were important for the subsequent growth of cucumber seedlings. Finding new ways to accelerate photoinduction without the decrease of biomass could effectively promote plant growth under fluctuating light [28]. Pre-treated with monochrome blue light could promote the growth of cucumber seedlings under subsequent fluctuating light, it was mainly because the increase of plant height, which increasing the light intensity on leaf surface, the increase of light intensity could significantly promote the growth of plants [40]. In addition, this result was similar to Kang et al. [27], who has found that acclimation to blue supplemental light at seedling stage could promote the growth of cucumber seedlings under sunlight through optimizing photosynthetic capacity and NPQ performance. The mechanism of red and blue light affecting the subsequent growth of cucumber seedlings still needs to be explored, for example, by comparing the hormone content and actual net photosynthetic rate during fluctuating light treatment.

## 4. Materials and Methods

### 4.1. Plant Materials and Experimental Design

The cucumber (*Cucumis sativus* L.) variety ‘Jinyou 365’ was used as test material, and this experiment was carried out in the phytotron at Institute of Urban Agriculture, Chinese Academy of Agricultural Sciences. First, cucumber seeds were soaked in deionized water at 25 °C for 6–8 h, then placed in an incubator with a temperature of 28 °C and a relative humidity of 80% for germination. After 20 h, seeds exhibiting uniform germination were sown in 72-hole plug trays (54.0 cm long × 30.0 cm wide × 5 cm deep, containing a 3:1:1 (*v*/*v*/*v*) mixture of peat, perlite, and vermiculite), while the temperature was 25 ± 2 °C, air relative humidity was 60%, light intensity was 100 μmol·m^−2^·s^−1^, light quality was white light, photoperiod was 14 h·d^−1^ (from 7:00 to 21:00), and carbon dioxide concentration was 400 ± 20 μmol·mol^−1^. After 7 d, when the cotyledon was fully expanded and the first true leaf was unfolded, uniform seedlings were selected and transplanted to a black container (80.0 cm long × 40.0 cm wide × 20 cm deep) for hydroponics. Each container contained 25 L Japan Yamazaki nutrient solution (NH_4_H_2_PO_4_ 0.5 mmol·L^−1^, Ca(NO_3_)_2_·4H_2_O 2.0 mmol·L^−1^, KNO_3_ 3.2 mmol·L^−1^, MgSO_4_·7H_2_O 1.0 mmol·L^−1^, with full-strength trace elements) in which 20 seedlings were planted. The pH of nutrient solution was 5.8–6.2, the electrical conductivity was 2.0–2.3 ms·cm^−1^, and oxygen was added with an air pump. The nutrient solution was replaced every 3 d.

Eight treatments were set in this study, including white light (W), monochrome red light (R), 90% red and 10% blue mixed light (9R1B), 70% red and 30% blue mixed light (7R3B), 50% red and 50% blue mixed light (5R5B), 30% red and 70% blue mixed light (3R7B), 10% red and 90% blue mixed light (1R9B) and monochromatic blue light (B). The maximum wavelength of red light was 660 nm, and the maximum wavelength of blue light was 450 nm. The spectral characteristics are shown in Figure 9; the light intensity was 100 μmol·m^−2^·s^−1^, which was the same as in previous studies on cucumber seedlings [4,34], and other environmental factors were same at the seedling cultivation stage. The shading cloth was used to separate each treatment. A ‘Heliospectra DYNA’ LED light (Heliospectra, Sweden) was used as a light source. Detailed information regarding this type of light can be found at ‘https://heliospectra.com/led-grow-lights/dyna/’ (accessed on 23 May 2024).

After 9 d of treatment, when the first true leaf was fully expanded, the growth and biochemical parameters were measured on the first true leaf. The first true leaf was collected, wrapped with aluminum foil, and quickly immersed in liquid nitrogen for quick freezing, then stored at −80 °C. In addition, 10 seedlings with uniform growth conditions were selected and treated with fluctuating light for 15 d. The fluctuation pattern was similar to previous study on Arabidopsis [28]. During fluctuating light treatment, white light intensity was quickly changed every 15 min between 30 μmol·m^−2^·s^−1^ and 300 μmol·m^−2^·s^−1^, the photoperiod was 10 h·d^−1^ (from 8:00 to 18:00), other environmental factors and management methods were not changed. All experiments have triply repeated independently with at least 3 plants used for measurement.

### 4.2. Growth and Morphology Parameter Measurement

Plant height and stem diameter were measured with vernier calipers. The plant height was the length between the stem base and the growth point; the stem diameter was measured at 0.5 cm below the cotyledon. A leaf area meter (LI-3100C, LI_COR Biosciences, Lincoln, NE, USA) was used to measure leaf area. The fresh weight of leaves, stems, and roots was measured using an electronic balance. For dry weight, after measuring fresh weight, seedlings were kept at 105 °C for 15 min and then kept at 80 °C until constant weight was reached to measure the dry weight. The specific leaf weight was calculated by specific leaf weight (g·cm^−2^) = leaf dry weight (g)/leaf area (cm^−2^) [33]. The root to shoot ratio was calculated by root to shoot ratio = root dry weight/shoot dry weight. The seedling index was calculated by seedling index = (stem diameter/plant height + root dry weight/shoot dry weight) × total dry weight [41]. The growth rate of cucumber seedlings during 15 days of fluctuating light treatment was calculated by growth rate = (growth parameters at 15 days − growth parameters at 0 days)/15.

### 4.3. Photosynthetic Gas Exchange and Chlorophyll Fluorescence Parameter Measurement

Photosynthetic gas exchange parameters under growth environment (actual photosynthesis) were measured by LI-6800 portable photosynthesis system (LI-COR Biosciences, Lincoln, NE, USA) equipped with a transparent leaf chamber. During the measurement, air flow rate was 500 μmol·s^−1^, CO_2_ concentration was 400 μmol·mol^−1^, air relative humidity was 60%, leaf temperature was 25 °C, and fan speed was 10000 rpm. Growth light was used as the light source. Intrinsic water use efficiency (iWUE) was calculated by A/Gs [37].

Photosynthetic gas exchange parameters during light induction were measured by LI-6800 equipped with the leaf chamber fluorometer. Light source was provided by a mixed red (80%, 635 nm) and blue (20%, 465 nm) LEDs on the top of the leaf chamber, while other environmental conditions were same as actual photosynthesis measurement. When measuring, data was recorded for 3 min in darkness, then light intensity quickly rose to 1500 μmol·m^−2^·s^−1^ and recorded for 50 min, at last, light intensity quickly dropped to 0 and recorded for 10 min. Data were recorded once per 10 s by an auto program.

Net photosynthetic rate in darkness (A_d_) and after light induction (A_f_), stomatal conductance in darkness (Gs_d_) and after light induction (Gs_f_), intercellular carbon dioxide concentration in darkness (Ci_d_) and after light induction (Ci_f_) could be directly recorded. Dark respiration rate was calculated by R_d_ = −A_d_.

Time to reach 50% steady-state net photosynthetic rate (T_50%A_) and 90% steady-state net photosynthetic rate (T_90%A_) were calculated by
$${IS}_{t}\left(\%\right)=\frac{A_{t}-A_{d}}{A_{f}-A_{d}}\times 100$$

A_t_ was the net photosynthetic rate at time t, A_d_ was the net photosynthetic rate in darkness, and A_f_ was the net photosynthetic rate after light induction. The corresponding time when A increases by 50% and 90% could also be calculated by this formula [10].

The calculation of time to reach 50% steady-state stomatal conductance (T_50%Gs_) and 90% steady-state stomatal conductance (T_90%Gs_) were similar to T_50%A_ and T_90%A_.

Rubisco activation rate (1/τ_R_) was calculated according to Liu et al. [22]. Firstly, net photosynthetic rate without stomatal restriction (A_t_*) was calculated by
$${A_{t}}^{*}=A_{t}\times \frac{{Ci}_{f}}{{Ci}_{t}}$$

A_t_ was the net photosynthetic rate at time t, Ci_f_ was the intercellular CO_2_ concentration after light induction, Ci_t_ was the intercellular CO_2_ concentration at time t.

Times after the light intensity increased was set as the X axis, the natural logarithm of the difference between net photosynthetic rate without stomatal restriction after light induction (A_f_*) and A_t_* (ln(A_f_* − A_t_*)) was set as the Y axis. As shown in Appendix A, the slope within 2 to 5 min was obtained by linear regression to calculate 1/τ_R_.

The lag time of stomatal conductance in response to the increase of light intensity (λ) was fitted by following formula [22]:$${Gs}_{t}=\left({Gs}_{f}-{Gs}_{d}\right)e^{-e^{\left(\frac{\lambda -t}{k}+1\right)}}+{Gs}_{d}$$

Gs_t_ was the stomatal conductance at time t, t was the time after the increase of light intensity, Gs_f_ was the stomatal conductance after light induction, Gs_d_ was the stomatal conductance in darkness, and k was a time constant. Origin was used for data fitting. During fitting, a custom function model was used, Gs_t_ was set as the dependent variable (y), t as the independent variable (x), (Gs_f_ − Gs_d_), and Gs_d_ were set as constants. λ and k were set as unknown parameters and obtained by fitting Gs_t_ and t (Appendix A).

The maximum increase rate of stomatal conductance during light induction (Sl_max_) was calculated by
$${Sl}_{max}=\frac{{Gs}_{f}-{Gs}_{d}}{k\times e}$$

The instantaneous stomatal limitation (LS) and biochemical limitation (LB) were calculated according to Xiong et al. [37]. Firstly, the net photosynthetic rate (A*) without stomatal limitation was calculated by
$$A^{*}=\frac{\left(A_{t}+R_{L})({Ci}_{f}-\Gamma^{*}\right)}{{Ci}_{t}-\Gamma^{*}}-R_{L}$$

A_t_ was the net photosynthetic rate at different time points, R_L_ was the dark respiration rate under light conditions, Ci_f_ was the intercellular carbon dioxide concentration when the light intensity is 1500 μmol·m^−2^·s^−1^, Γ* was the carbon dioxide compensation point at the chloroplast carboxylation site, and Ci_t_ was the intercellular carbon dioxide concentration at different times. Γ* and R_L_ were determined according to Xiong et al. [42], the results are shown in Appendix A and Appendix A.

Instantaneous stomatal limitation (LS) and biochemical limitation (LB) were calculated by
$$LS=\frac{A^{*}-A_{t}}{A_{f}+R_{L}}$$
$$LB=\frac{A_{f}-A^{*}}{A_{f}+R_{L}}$$

### 4.4. Stomatal Characteristic Measurement

First, the leaves were fixed and dehydrated according to [43]. Briefly, leaf discs were extracted from the same part of the leaves, then the leaf discs were quickly immersed in the fixing solution containing 2.5% (*w*/*v*) glutaraldehyde, and stored at −4 °C for 24 h. After fixation, the leaf discs were washed with PBS buffer and then gradient dehydration was carried out with different concentrations of ethanol (30%, 50%, 70%, 80%, 90%, 95%, and 100%). After dehydration, the abaxial epidermis of the leaves was removed with transparent tape and placed on the slide and then observed under a microscope (OLYMPUS IX73). For each slide, 5 fields of view and 15 stomata were selected for measurement. As the adaxial epidermis is difficult to tear, the stomata on the adaxial of leaves were observed by an imprinting method [27].

Stomata-related parameters were measured by Image J and calculated according to [18]: stomata density (SD) was the number of stomata per area. Epidermal cell density (RCD) was the number of cells except stomata per area. Stomatal index (SI) = number of stomata/(number of stomata + number of epidermal cells). Stomatal aperture area per area (SAPA) = average stomatal opening area × stomatal density. Stomatal area per area (SAPA) = (stomatal area × number of stomata in each field)/total field area × 100. Epidermal cell area (ECA) = (field area − total stomatal area)/number of epidermal cells.

### 4.5. Statistics

The mean value and standard deviation (SD) were calculated using Excel 2016 (Microsoft Corporation, USA). Data normality and variance homogeneity were tested by SPSS 16.0 (IBM, New York, NY, USA) and then Duncan’s multiple range test was used to compare the differences among different treatments (*p* < 0.05). Origin 2021 (OriginLab Corp., Northampton, MA, USA) was used to draw pictures and the Correlation Plot App was used to analyze the correlation between different indicators according to the Pearson correlation coefficient with the two-tailed test.

## 5. Conclusions

Compared with red light, blue light increases maximum photosynthetic capacity and photoinduction rate of cucumber seedlings, but decreases the biomass because of low leaf area and actual net photosynthetic rate, cucumber seedlings growing under 9R1B exhibit the highest biomass and actual net photosynthetic rate. The higher maximum photosynthetic capacity and photoinduction rate does not corresponding to the high growth rate under subsequent fluctuating light. On the contrary, seedlings pre-treated with 9R1B and B grow better under subsequent fluctuating light because of higher plant height and leaf area, indicating plant morphology and actual photosynthetic rate play key roles in promoting plant growth. In summary, when the light intensity is 100 μmol·m^−2^·s^−1^, 9R1B is more suitable for the cultivation of cucumber seedlings.

## Figures and Tables

**Figure 1 plants-13-01668-f001:**
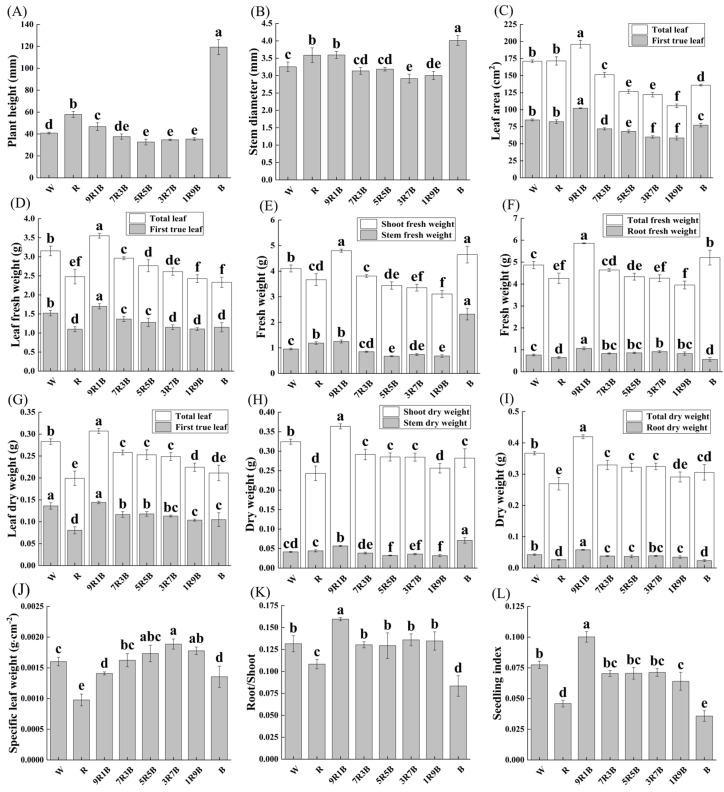
Effects of red and blue light on the growth of cucumber seedlings. (**A**) Plant height, (**B**) stem diameter, (**C**) area of total leaf and first true leaf, (**D**) fresh weight of total leaf and first true leaf, (**E**) fresh weight of shoot and stem, (**F**) fresh weight of total plant and root, (**G**) dry weight of total leaf and first true leaf, (**H**) dry weight of shoot and stem, (**I**) dry weight of total plant and root, (**J**) specific leaf weight, (**K**) root–shoot ratio (Root/Shoot), (**L**) seedling index. Different letters indicate significant differences among each treatment (*p* < 0.05). White light (W), monochrome red light (R), 90% red and 10% blue mixed light (9R1B), 70% red and 30% blue mixed light (7R3B), 50% red and 50% blue mixed light (5R5B), 30% red and 70% blue mixed light (3R7B), 10% red and 90% blue mixed light (1R9B), and monochromatic blue light (B).

**Figure 2 plants-13-01668-f002:**
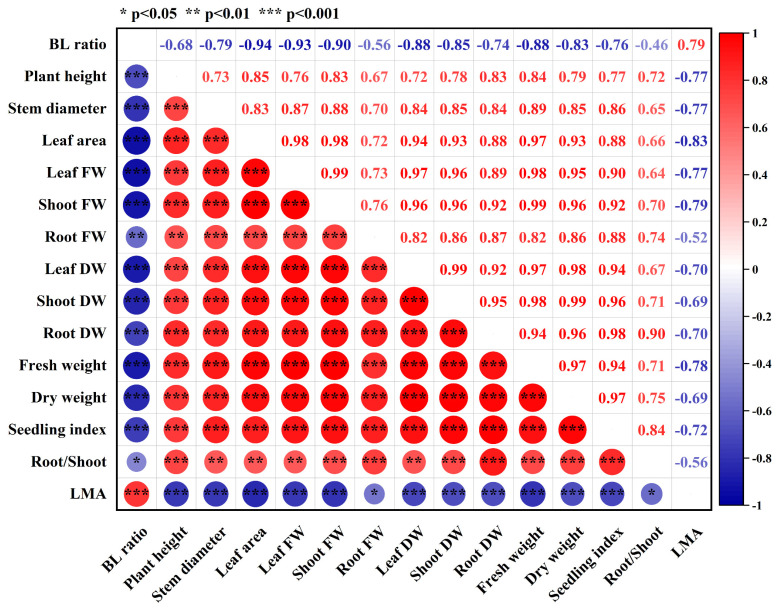
Correlation analysis between plant growth parameters under mixed red and blue light. BL ratio: blue light ratio, FW: fresh weight, DW: dry weight, Root/Shoot: root–shoot ratio, LMA: specific leaf weight. Red color and positive values represent the positive correlation between two parameters, while blue color and negative values represent the negative correlation between two parameters. ‘*’ represents the significant correlation between two parameters at *p* < 0.05, while ‘**’ at *p* < 0.01, and ‘***’ at *p* < 0.001.

**Figure 3 plants-13-01668-f003:**
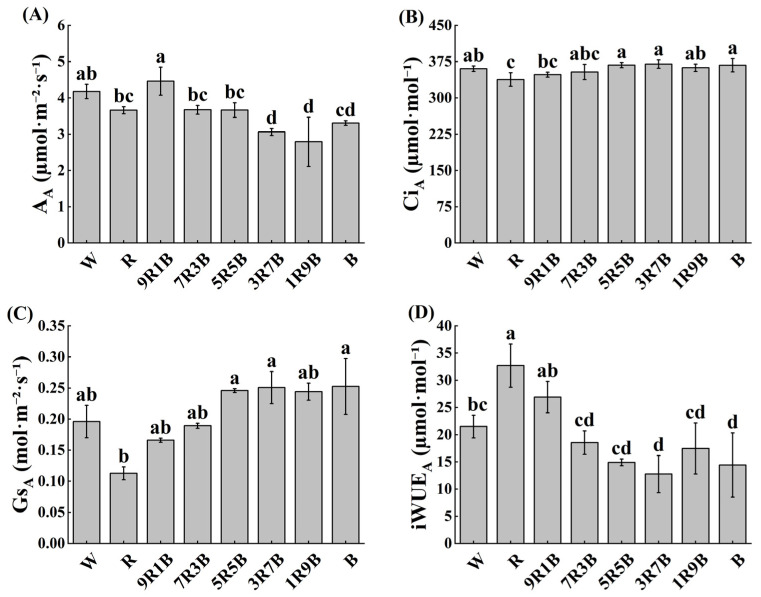
Effects of red and blue light on the actual photosynthetic characteristics of cucumber seedlings. (**A**) Actual net photosynthetic rate (A_A_), (**B**) actual intercellular carbon dioxide concentration (Ci_A_), (**C**) actual stomatal conductance (Gs_A_), (**D**) actual intrinsic water use efficiency (iWUE_A_). Different letters indicate significant differences among each treatment (*p* < 0.05).

**Figure 4 plants-13-01668-f004:**
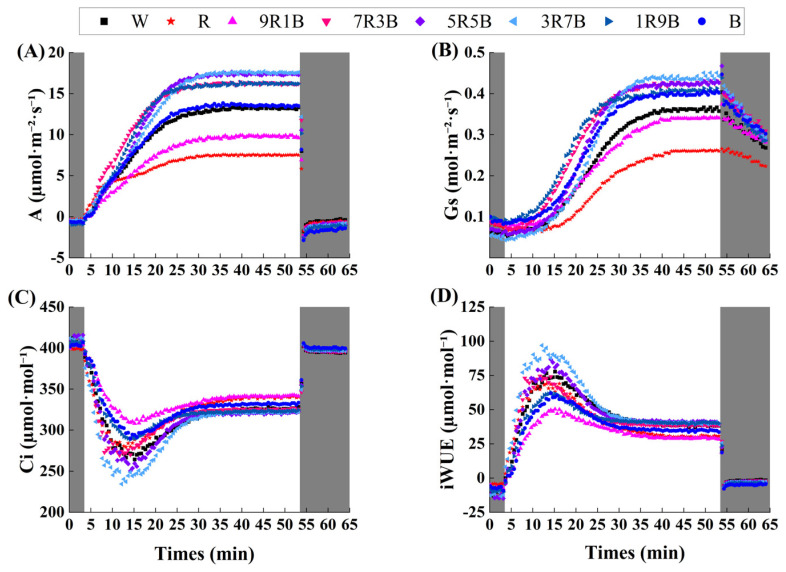
Variations of photosynthetic gas exchange parameters after light induction. (**A**) Net photosynthetic rate (A), (**B**) stomatal conductance (Gs), (**C**) intercellular carbon dioxide concentration (Ci), (**D**) intrinsic water use efficiency (iWUE). The dark gray area represents a light intensity of 0, and the white area represents a light intensity of 1500 μmol·m^−2^·s^−1^.

**Figure 5 plants-13-01668-f005:**
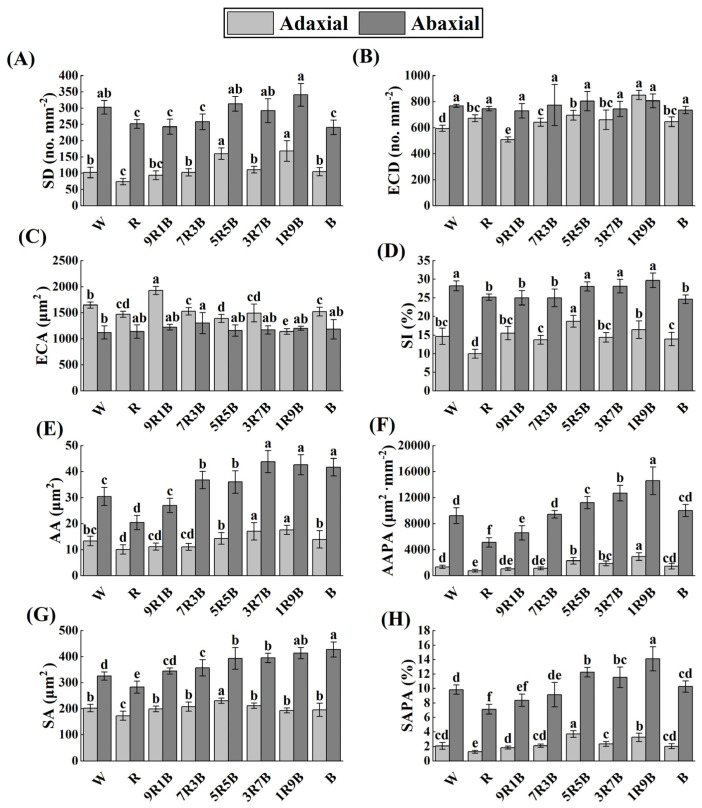
Effect of red and blue light on stomatal characteristics of cucumber seedlings. (**A**) Stomatal density (SD), (**B**) epidermal cell density (ECD), (**C**) epidermal cell area (ECA), (**D**) stomatal index (SI), (**E**) stomatal aperture area (AA), (**F**) stomatal aperture area per area (AAPA), (**G**) stomatal area (SA), (**H**) stomatal area per area (SAPA). The difference analysis was individually conducted on different leaf sides. Different letters indicate significant differences among each treatment (*p* < 0.05).

**Figure 6 plants-13-01668-f006:**
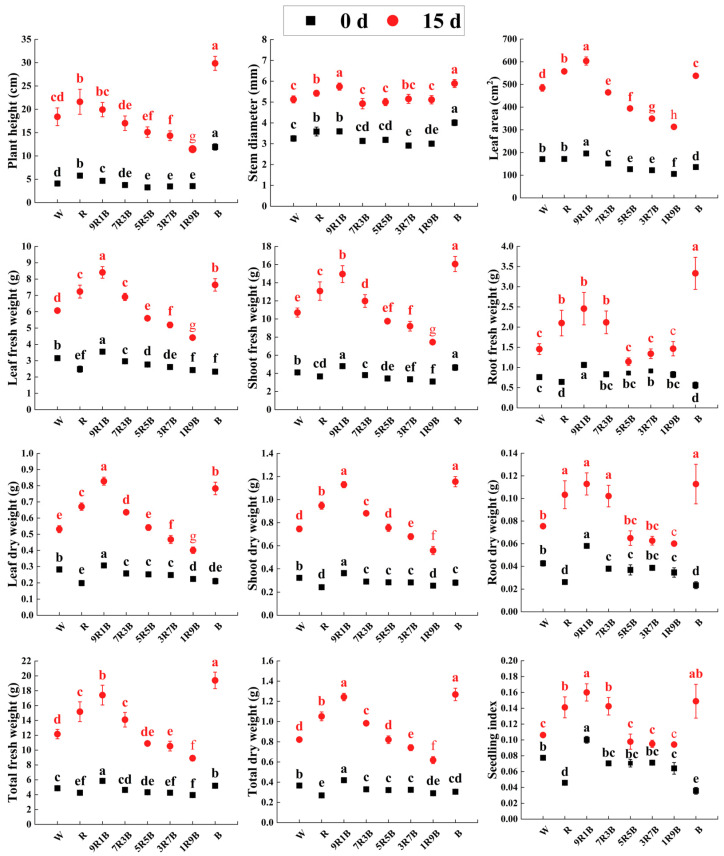
Growth of cucumber seedlings under subsequent fluctuating light for 0 d and 15 d. The difference analysis was individually conducted under different days. Different letters indicate significant differences among each treatment (*p* < 0.05).

**Figure 7 plants-13-01668-f007:**
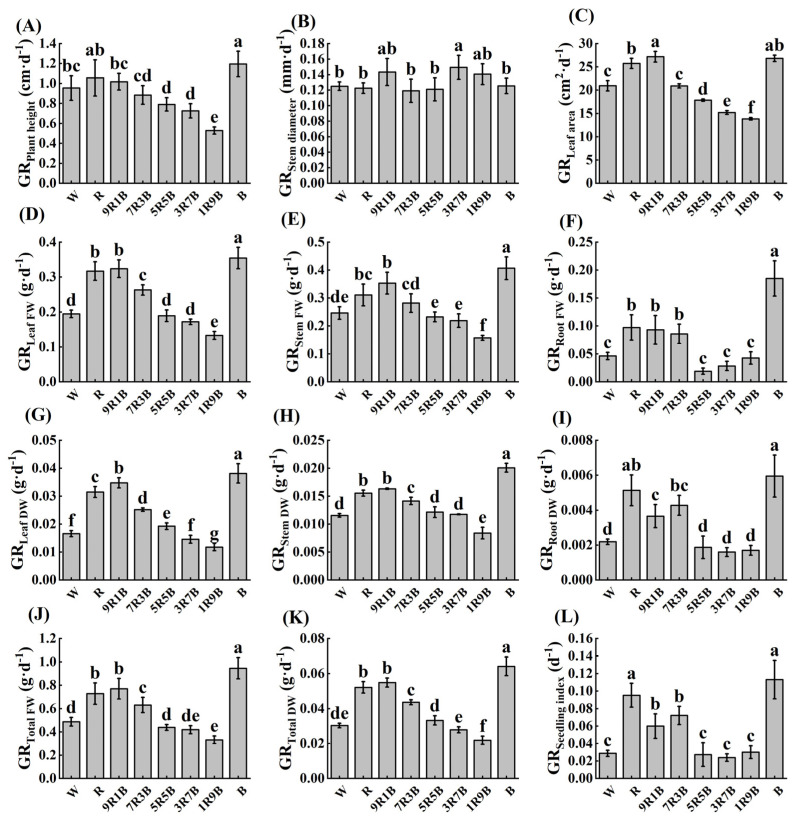
Growth rate of cucumber seedlings under subsequent fluctuating light. (**A**) Growth rate of plant height (GR_plant height_), (**B**) growth rate of stem diameter (GR_stem diameter_), (**C**) growth rate of leaf area (GR_leaf area_), (**D**) growth rate of leaf fresh weight (GR_leaf FW_), (**E**) growth rate of stem fresh weight (GR_stem FW_), (**F**) growth rate of root fresh weight (GR_root FW_), (**G**) growth rate of leaf fresh weight (GR_leaf DW_), (**H**) growth rate of stem dry weight (GR_stem DW_), (**I**) growth rate of root dry weight (GR_root DW_), (**J**) growth rate of total fresh weight (GR_total FW_), (**K**) growth rate of total dry weight (GR_total DW_), (**L**) growth rate of seedling index(GR_seedling index_). Different letters indicate significant differences among each treatment (*p* < 0.05).

**Figure 8 plants-13-01668-f008:**
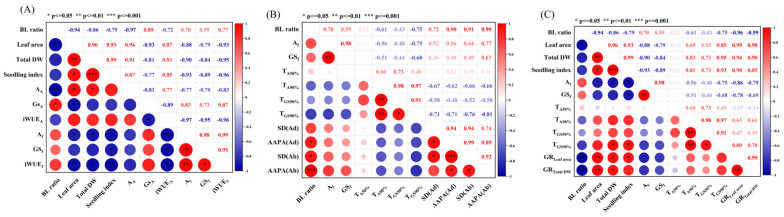
Correlation of key parameters under mixed red and blue light. (**A**) Correlation between plant growth and photosynthesis parameters under mixed red and blue light; (**B**) correlation between steady-state photosynthetic characteristics, dynamic photosynthetic characteristics, and stomatal characteristics; (**C**) correlation between biomass, photosynthesis parameters, and growth rate. Red color and positive value represent the positive correlation between two parameters, while blue color and negative value represent the negative correlation between two parameters. ‘*’ represents the significant correlation between two parameters at *p* < 0.05, while ‘**’ at *p* < 0.01, and ‘***’ at *p* < 0.001.

**Figure 9 plants-13-01668-f009:**
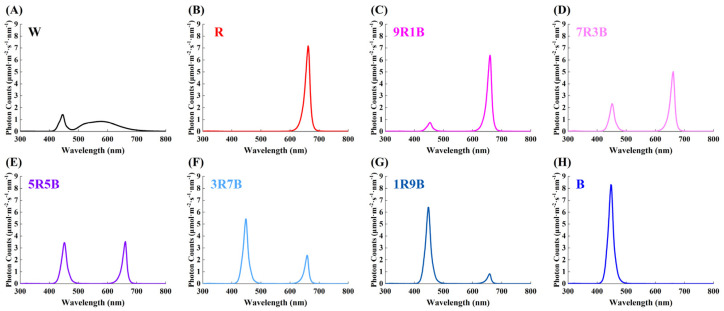
Light spectra of different treatments. (**A**) White light (W), (**B**) monochrome red light (R), (**C**) 90% red and 10% blue mixed light (9R1B), (**D**) 70% red and 30% blue mixed light (7R3B), (**E**) 50% red and 50% blue mixed light (5R5B), (**F**) 30% red and 70% blue mixed light (3R7B), (**G**) 10% red and 90% blue mixed light (1R9B) and (**H**) monochromatic blue light (B).

**Table 1 plants-13-01668-t001:** Effects of red and blue light on steady-state photosynthetic parameters of cucumber seedlings.

Treatments	A_f_	Gs_f_	Ci_f_	iWUE_f_	Gs_d_	R_d_
	(µmol·m^−2^·s^−1^)	(mol·m^−2^·s^−1^)	(µmol·mol^−1^)	(µmol·mol^−1^)	(mol·m^−2^·s^−1^)	(µmol·m^−2^·s^−1^)
W	13.20 ± 0.31 b	0.36 ± 0.07 a	326.79 ± 12.70 b	37.95 ± 7.97 a	0.064 ± 0.025 a	0.65 ± 0.13 ab
R	7.43 ± 0.34 d	0.26 ± 0.06 b	341.64 ± 9.36 a	29.22 ± 5.83 b	0.073 ± 0.017 a	0.25 ± 0.15 c
9R1B	9.72 ± 0.89 c	0.34 ± 0.01 ab	341.99 ± 2.85 a	28.57 ± 1.50 b	0.082 ± 0.007 a	0.58 ± 0.12 ab
7R3B	16.17 ± 1.13 a	0.42 ± 0.05 a	325.70 ± 3.15 b	38.28 ± 1.94 a	0.079 ± 0.017 a	0.75 ± 0.17 ab
5R5B	17.33 ± 0.72 a	0.42 ± 0.01 a	321.29 ± 2.55 b	40.85 ± 1.58 a	0.063 ± 0.039 a	0.60 ± 0.13 ab
3R7B	17.53 ± 0.27 a	0.43 ± 0.04 a	321.17 ± 5.13 b	40.63 ± 3.37 a	0.053 ± 0.029 a	0.44 ± 0.12 bc
1R9B	16.14 ± 1.53 a	0.41 ± 0.04 a	323.17 ± 9.81 b	40.07 ± 5.68 a	0.089 ± 0.027 a	0.79 ± 0.28 a
B	13.47 ± 0.97 b	0.40 ± 0.07 a	332.21 ± 7.92 ab	34.33 ± 4.67 ab	0.082 ± 0.012 a	0.64 ± 0.13 ab

Net photosynthetic rate after light induction (A_f_), stomatal conductance after light induction (Gs_f_), intercellular carbon dioxide concentration after light induction (Ci_f_), intrinsic water use efficiency after light induction (iWUE_f_), stomatal conductance in darkness (Gs_d_), dark respiration rate (R_d_). Data are presented as means ± SE (n = 3). Different letters indicate significant differences among each treatment (*p* < 0.05).

**Table 2 plants-13-01668-t002:** Effects of red and blue light on dynamic photosynthetic parameters of cucumber seedlings.

Treatments	T_50%A_	T_90%A_	1/τ_R_	T_50%Gs_	T_90%Gs_	λ	Sl_max_
	(min)	(min)	(min^−1^)	(min)	(min)	(min)	(mol·m^−2^·s^−1^)
W	13.56 ± 0.10 ab	23.33 ± 0.44 abc	0.085 ± 0.015 c	24.72 ± 2.50 a	33.94 ± 3.62 ab	9.85 ± 2.82 b	0.018 ± 0.008 b
R	8.22 ± 2.74 c	26.17 ± 1.30 a	0.290 ± 0.161 a	23.78 ± 1.75 a	37.00 ± 3.33 a	13.68 ± 0.97 a	0.012 ± 0.006 c
9R1B	13.94 ± 1.02 ab	24.50 ± 2.89 ab	0.074 ± 0.008 c	25.83 ± 3.88 a	34.83 ± 2.68 ab	10.96 ± 0.93 ab	0.015 ± 0.004 bc
7R3B	11.39 ± 0.19 bc	19.83 ± 2.52 cd	0.155 ± 0.023 b	21.28 ± 1.00 a	28.67 ± 1.09 bc	8.57 ± 2.77 bc	0.024 ± 0.003 ab
5R5B	14.56 ± 1.39 ab	21.22 ± 0.59 bcd	0.074 ± 0.009 c	23.00 ± 1.48 a	29.50 ± 2.35 bc	10.44 ± 1.83 ab	0.028 ± 0.008 a
3R7B	15.56 ± 2.59 a	22.94 ± 1.54 abc	0.085 ± 0.034 c	25.28 ± 2.61 a	30.78 ± 2.71 bc	11.85 ± 1.63 ab	0.028 ± 0.006 a
1R9B	12.94 ± 1.07 ab	18.11 ± 2.69 d	0.077 ± 0.006 c	20.67 ± 2.78 a	26.22 ± 5.22 c	7.78 ± 1.65 c	0.025 ± 0.004 ab
B	13.83 ± 3.59 ab	22.11 ± 2.18 bc	0.101 ± 0.060 bc	22.44 ± 4.40 a	29.72 ± 3.56 bc	10.66 ± 2.70 ab	0.023 ± 0.005 ab

Time to reach 50% steady-state net photosynthetic rate (T_50%A_) and 90% steady-state net photosynthetic rate (T_90%A_), Rubisco activation rate (1/τ_R_), time to reach 50% steady-state stomatal conductance (T_50%Gs_) and 90% steady-state stomatal conductance (T_90%Gs_), lag time of stomatal conductance in response to the increase of light intensity (λ), the maximum increase rate of stomatal conductance during light induction (Sl_max_). Different letters indicate significant differences among each treatment (*p* < 0.05).

## Data Availability

Data are contained within the article and Appendix A.

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
