# Peer review of "Effects of Red and Blue Light on the Growth, Photosynthesis, and Subsequent Growth under Fluctuating Light of Cucumber Seedlings"

_plants, 2024, doi:10.3390/plants13121668_

Round 1

Reviewer 1 Report

Comments and Suggestions for Authors

I attached the file. Please refer to the file.

Comments on the Quality of English Language

Over all good, but there are some typo mistakes. Please check it.

Author Response

Thanks for your comments and we have revised the manuscript according to your commments. Please see the attachment.

Reviewer 2 Report

Comments and Suggestions for Authors

The manuscript examines the effects of red and blue light on cucumber seedling's growth, steady-state photosynthesis, dynamic photosynthesis, and subsequent growth under fluctuating light. The manuscript has been prepared properly, but there are some editorial comments:

Letters in Figures 1, 3, 5, 7 indicating significant differences could be slightly smaller, especially in Figure 5 - they cover the columns or each other in some places. Letters in Figures 1, 3, 5, and 7 indicating significant differences could be slightly smaller, especially in Figure 5 - they cover the columns or each other in some places. In Figure 9, the data could be presented more clearly. Maybe it is possible to provide a separate small graphic for an individual variant.

Author Response

(The authors gave the same response as above.)

Reviewer 3 Report

Comments and Suggestions for Authors

Dear authors,

The article is interesting but needs some improvement.

The abstract and introduction are well-written.

Results section

1. How would you explain your results regarding "B" effect on plant height and stem thickness? In the introduction, you mention that B inhibits height growth but increases stem diameter (lines 43, 44). In your results, "B" stimulates both metrics (Figure 1A, B). What caused this?

2. The description of the results obtained needs to be clarified. Short sentences are nice but sometimes insufficient (lines 138, 143, 191, 194, and all others).

"AA reached the lowest under 1R9B and reached the highest under 9R1B."

"GsA reached the lowest under R and reached the highest under B."

"T50%A reached the lowest under R and reached the highest under 3R7B."

" T90%A reached the lowest under 1R9B and reached the highest under R."

3. When expressing results in %, two decimal places are not required. Usually, the change in the second character after the comma is of no particular importance. You can only comment on the statistically significant results to simplify the text. The rest can be seen in the graphs.

4. How and when was dark respiration measured?

5. How will you explain the inverse relationship in the time to reach 50% of the steady-state net photosynthetic rate (T50%A) and 90% of the steady-state net photosynthetic rate (T90%A) for R treatment? (Table 2)

Discussion

Please rewrite the sentence. When you refer to someone directly, it is good to write their name and, in the brackets give the number under which they are cited in the literature. Line 354:

"The result was similar to [5], which has found that monochrome red light could inhibit the growth of cucumber and tomato."

Material and methods

Isn't the light intensity too low? Usually, cucumbers are grown under controlled conditions at 300-400 micromolar light intensity to avoid etiolation.

Please provide lamp specifications.

The conclusion corresponds to the results obtained.

Literary sources are used correctly.

Comments on the Quality of English Language

The article needs to be reviewed by a native English speaker.

Author Response

(The authors gave the same response as above.)

Reviewer 4 Report

Comments and Suggestions for Authors

The manuscript is devoted to the study of the influence of light of different quality on the initial period of growth of Cucumis sativus L. cv. Jinyou 365. A very detailed characterization of growth, many morphological indices, photosynthetic function is evaluated and characterization of stomata is given. The work is large and has been done very carefully. The effects of different light quality have been studied for a long time and the results have been published in many articles. The great advantage of the paper is the use of not only white, monochromatic red and blue light, but also their different combinations. The results have some theoretical interest, but are mainly aimed at optimizing the cultivation of cucumbers in greenhouse conditions. In this respect they may prove useful. In any case, the optimal combination of red and blue light for cucumber cultivation has been chosen and these results can already be applied in practice somewhere.

However, it is not clear to me why dynamic illumination after the action of light of different quality was studied. Fluctuation of light intensity from 30 μmol-m-2-s-1 and 300 μmol-m-2-s-1 is impossible to imagine in nature. In artificial illumination such a situation will never happen either.  What this is being studied for. It would be good to focus a little bit on this point. I didn't find in the methods how long this dynamic lighting was used, which is also interesting to know.

 At least it's not clear to me why taller plants would tolerate dynamic lighting more easily. Light of this quality and mode will affect both low and taller plants equally.

 Cucumbers are not grown for more biomass, but for the cucumbers themselves. Will the optimized lighting regimes affect the yield. Maybe this work in the sense of obtaining increased yield is of little importance.

 I would be grateful if the authors would take into account my comments and make some minor revisions to the manuscript.

 Overall, the manuscript is good and I would recommend that it be accepted for publication in the journal with minor revisions.

Author Response

(The authors gave the same response as above.)

Reviewer 5 Report

Comments and Suggestions for Authors

The study, “Effects of red and blue light on the growth, photosynthesis, and subsequent growth under fluctuating light of cucumber seedlings,” sheds insight into the plant performance in various lighting conditions. The study appears meticulously planned, and the results are explained in detail. As per the authors, the results indicate that red and blue light significantly affected plant growth, morphological characteristics, and the photosynthetic ability of cucumber seedlings. Since it is well-known that blue light increases maximum photosynthetic capacity and photoinduction rate in plants, it appears that the present study does not bring any novelty, specifically considering that the cucumber fruit is of significant importance. It requires months to get the fruits. Therefore, the present study should have been supported by some experiments of later-stage plants bearing fruits and the effects of lighting conditions on fruits. Cucumber leaves and fast-growing fruits belong to the source and sink organs. Some experiments related to source-sink could have been added. These experiments may provide exciting results. I hope the authors might be working towards some of these experiments to support current findings.  

Comments on the Quality of English Language

Acceptable

Author Response

(The authors gave the same response as above.)
